# Evolutionary age correlates with range size across plants and animals

Adriana Alzate [1,2,3,4] ✉, Roberto Rozzi [1,5,6], Julian A. Velasco [7], D. Ross Robertson [8], Alexander Zizka[9], Joseph A. Tobias [10], Adrian Hill[11,12], Christine D. Bacon[11,12], Thijs Janzen [13], Loïc Pellissier [14,15], Fons van der Plas [16], James Rosindell [10] & Renske E. Onstein [1,2,4]

More than 40 thousand species of plants and animals are facing extinction worldwide. Range size is one of the strongest determinants of extinction risk, but the causes underlying the wide variation in natural range sizes remain poorly understood. Here, we investigate how species' age is related to present-day range size for over 26,000 species of mammals, birds, reptiles, amphibians, reef fishes, and plants. We show that, on average, older species have larger ranges across all groups except for marine mammals, but the strength of the age-range size relationship depends on taxonomic scale. Furthermore, while our results confirm the well-established pattern of smaller range sizes for species restricted to islands (compared to mainland) or with limited dispersal abilities (compared to good dispersers), we show that the correlation between species age and range size is stronger in these groups, suggesting that island dynamics and dispersal ability modulate this relationship. Our study reveals that species with small ranges, and thus increased extinction risk, tend to be restricted to islands, are poor dispersers, or have recently evolved.

What influences a species' range size? This is a long-standing question in macroecology and biogeography with strong implications for conservation. Species with narrow geographical ranges face a higher risk of extinction compared to widespread species[1–4]. Narrow-ranged species tend to have lower overall abundance and smaller local populations[5–7], making them vulnerable to environmental perturbations that result in local extinction[2]. Consequently, understanding how ecological and evolutionary factors affect species' range size can support our understanding of species vulnerabilities and support the establishment of global conservation priorities.

A potential driver of species range size variation is 'species evolutionary age' (age hereafter). Older lineages are expected to have obtained larger distributional extents than younger species, because they have had more time for range expansion after speciation (the 'age and area model'[8]). Even though species ranges may fluctuate over ecological time scales due to demographic processes and local source-

[1]German Centre for Integrative Biodiversity Research (iDiv) Halle - Jena - Leipzig, Leipzig, Germany. [2]Leipzig University, Leipzig, Germany. [3]Aquaculture and Fisheries Group, Wageningen University and Research, Wageningen, The Netherlands. [4]Naturalis Biodiversity Center, Leiden, the Netherlands. [5]Central Repository of Natural Science Collections (ZNS), Martin Luther University Halle-Wittenberg, Halle (Saale), Germany. [6]Museum für Naturkunde, Leibniz Institute for Evolution and Biodiversity Science, Berlin, Germany. [7]Instituto de Ciencias de la Atmósfera y Cambio Climático, Universidad Nacional Autónoma de México, Mexico City, Mexico. [8]Smithsonian Tropical Research Institute, Balboa, Panama. [9]Department of Biology, Philipps-University Marburg, Marburg, Germany. [10]Department of Life Sciences, Imperial College London, Ascot, United Kingdom. [11]Department of Biological and Environmental Sciences, University of Gothenburg, Gothenburg, Sweden. [12]Gothenburg Global Biodiversity Centre, Gothenburg, Sweden. [13]Groningen Institute for Evolutionary Life Sciences, University of Groningen, Groningen, the Netherlands. [14]Ecosystems & Landscape Evolution, Department of Environmental Systems Science, Institute of Terrestrial Ecosystems, Swiss Federal Institute of Technology, ETH Zurich, Zurich, Switzerland. [15]Department of Landscape Dynamics & Ecology, Swiss Federal Institute for Forest, Snow and Landscape Research WSL, Birmensdorf, Switzerland. [16]Plant Ecology and Nature Conservation, Wageningen University and Research, Wageningen, The Netherlands. ✉e-mail: adria.alzate@gmail.com

sink dynamics, distributional extents widen over longer (e.g., million-year) time scales[9]. Despite the expected link between age and range size, previous studies provide conflicting results[10–13]. Support for a positive effect of age on range size is observed in fossil molluscs and trilobites[14,15], but extant organisms show a more complex picture (Supplementary Table 1). These previous studies vary strongly in their spatial extent (e.g., focusing on regional versus global ranges), taxonomic scale[16] (e.g., focusing on variation within genera, families, or orders), and methodological aspects (e.g., the method used to quantify range size). Furthermore, a myriad of factors other than species age influence range size[17], limiting our understanding of a possible general effect of age on range size[10].

Besides species age, species range size is influenced by a wide range of ecological, evolutionary and geographical factors that determine the rate by which species expand their range, the variety of habitat types that a species can persist in ('niche breadth'), and colonization-extinction dynamics[17]. For example, the available 'niche' space (e.g., island or habitat availability or size)[18] influences the

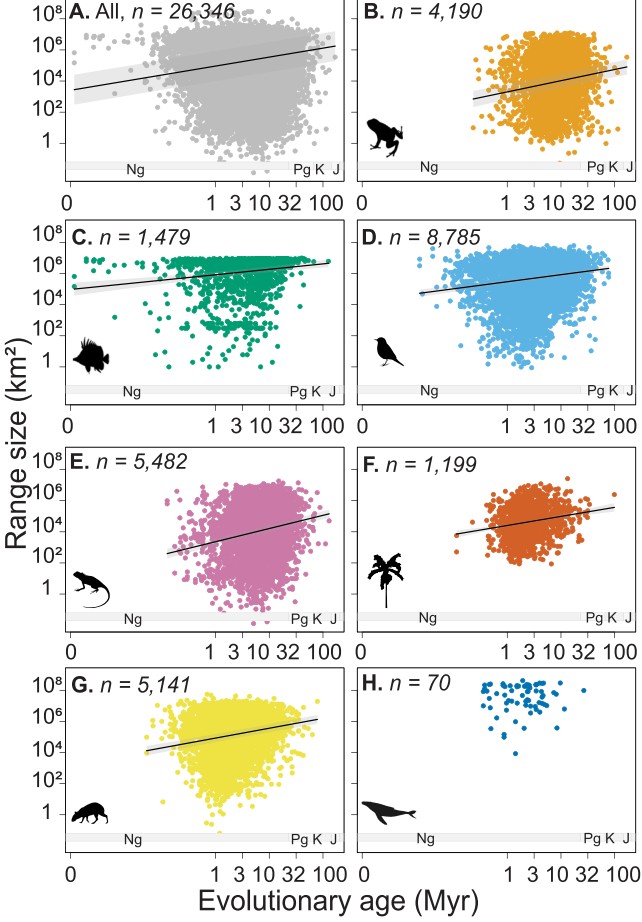

**Fig. 1 | The relationship between species age and range size.** Relationship between species age (median age across 100 phylogenetic trees) and range size, with the solid line representing the predicted mean (model fit) and the shaded area representing the 95% confidence interval of the predictions, for **A** all species, and (**B**–**H**) separated for the seven broad taxonomic groups: (**B**) amphibians, **C** reef fishes, **D** birds, **E** reptiles, **F**) palms, **G** terrestrial mammals, and H) marine mammals. Geological periods are denoted as *Ng* Neogene, *Pg* Paleogene, *K* Cretaceous, and *J* Jurassic. Note that both axes are log₁₀-transformed, consistent with the transformation used in the analyses. *n* number of species; *Myr* million years. These results were robust to outliers and different approaches to calculating species age (Supplementary Fig. 2–3, and Supplementary Table 3–4). Source data is provided as a Source Data file.

maximum range extent, whereas dispersal ability influences the rate of range expansion[19–22]. Good dispersers (i.e., species that can move easily across barriers and/or over great distances) may attain large ranges faster than less dispersive species[22]. Therefore, good dispersers might have larger ranges than expected based on age only. Moreover, range size dynamics are likely species and lineage-specific[10,16], with less dispersive or narrow-ranged species and lineages (e.g., amphibians) showing more pronounced age-range size relationships than more dispersive or wide-ranging species and lineages (e.g., marine mammals). Disentangling the effect of species age on range size thus requires accounting for the geographical, ecological, and taxonomic factors that modulate the age- range size relationship.

Here, we determine the relationship between species age and range size for over 26,000 species from across the Tree of Life, including amphibians, reef fishes, birds, reptiles, palms, terrestrial and marine mammals. We hypothesize that species age has a positive effect on range size. The strength of this effect may depend on taxonomic scale, geographical context and species' dispersal abilities. First, we expect the relationship between age and range size to be more pronounced at large taxonomic scales (e.g., class, orders) than at narrow taxonomic scales (e.g., families, genera). Range size and/or age variability tend to be lower at smaller taxonomic scales consisting of closely related species that share similar ages, distributions, and dispersal abilities, leading to no or weak age-range relationships. Second, we hypothesize that the relationship between age and range size is less pronounced or absent on islands than on mainland, because the maximum range size that endemic species can attain on islands is geographically rather than age-constrained. Third, we expect that the relationship between age and range size is more pronounced for less dispersive than highly dispersive species, because dispersive species may attain maximum range sizes faster than less dispersive species, reducing the effect of age on range size. Hence, dispersive species may have larger ranges, and less dispersive species smaller ranges, than expected based on age only.

Our results show a positive correlation between species age and range size, but this relationship is influenced by taxonomic scale, geographical context, and the species' dispersal abilities. Our study reveals that species with small ranges, and thus increased extinction risk, tend to be restricted to islands, are poor dispersers, or have evolved recently. Understanding the eco-evolutionary dynamics that shape species' range size is crucial for predicting species' vulnerability to extinction, especially in the context of changing environmental conditions and the need for targeted conservation efforts.

## Results

### Species age is positively correlated with range size

We determined the effect of age on range size for 26,345 lineages of seven major groups of animals and plants (Supplementary Fig. 1), using linear mixed-effects models. As a random effect, 'family' was nested within 'order', which was nested within the 'broad taxonomic group'. We found that age is significantly positively related to range size ($Z = 0.16$, SE = 0.006, $t(25,380) = 25.49$, $p < 0.0001$; Fig. 1A). This effect is significant for all broad taxonomic groups ($Z = 0.15$-$0.20$), except for marine mammals, which showed no significant effect (Fig. 1B–H, and Supplementary Table 2). These results were robust to outliers (age values deviating more than three standard deviations from the mean, Supplementary Fig. 2, Supplementary Table 3) and different approaches of calculating species age[23] (e.g., when adjusting for extinction probability, Supplementary Table 4 and Supplementary Fig. 3). Finally, null models (1000 randomizations of age across taxa) that break the phylogenetic signal of age supported our findings by showing that the observed effects (i.e., standardized effect sizes) of age on range size strongly deviated from a null expectation of no effect of age on range size for all taxa, except for marine mammals (Fig. 2).

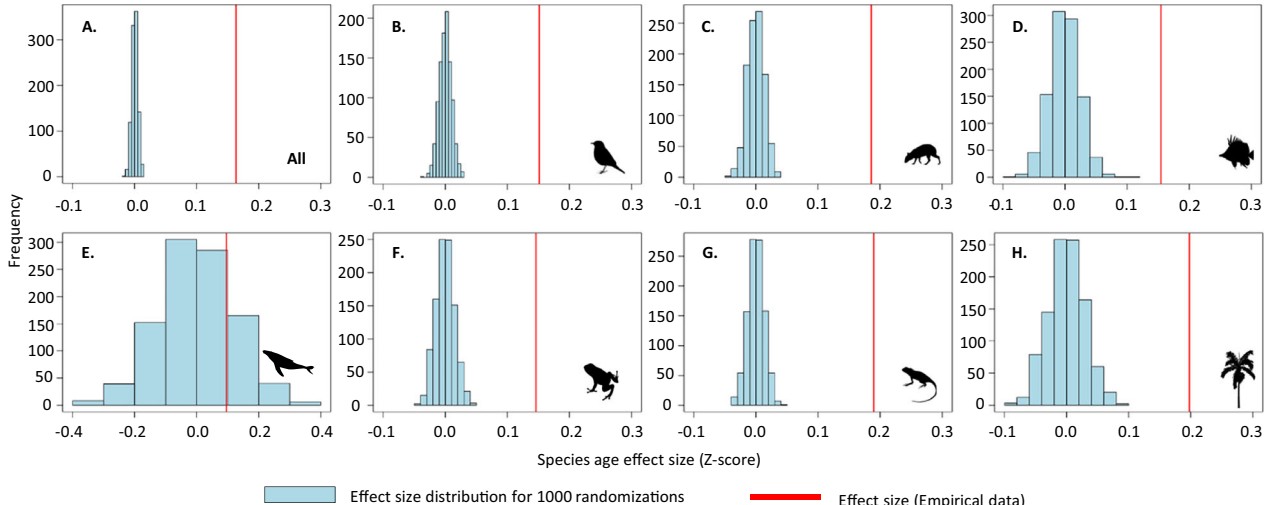

**Fig. 2 | The empirical relationship between species age and range size significantly deviates from the null expectation.** The null scenario assumes no correlation between species age and range size, indicating an absence of phylogenetic signal for age. Frequency distribution of effect sizes (indicated by blue bars) from linear mixed-effect models based on 1000 randomizations of age across the phylogeny, for (**A**) all taxa combined and (**B**–**H**) separated for the seven broad taxonomic groups: **B** amphibians, **C** reef fishes, **D** birds, **E** reptiles, **F** palms, **G** terrestrial mammals, and H) marine mammals. The red line denotes the observed effect size of age on range size for the empirical data. Source data is provided as a Source Data file.

## The age-range size relationship varies across taxonomic scales

We determined the strength of the age-range size relationship at three taxonomic scales, by running separate linear models: broad taxonomic scale (amphibians, reef fishes, birds, reptiles, palms, terrestrial and marine mammals), order scale (108 orders), and family scale (418 families). To test the relationship between age and range size across these scales, while accounting for sampling variance, we ran three meta-analyses, including all the individual linear model estimates of age on range size within each scale. We found scale-dependent relationships between age and range size. Our meta-analysis confirmed a positive relationship between age and range size across all taxonomic scales (Figs. 3a, b). The effect was strongest at the broad taxonomic scale (K = 7, Estimate = 0.17 [0.16 - 0.18], $p < 0.0001$; test for heterogeneity: Q(df = 6) = 6.54, $p = 0.37$), followed by the order level (K = 86, Estimate = 0.16 [0.14 - 0.18], $p < 0.0001$; test for heterogeneity: Q (df = 85) = 640.91, $p < 0.0001$). At the family level, the effect was weakest and exhibited high variability and uncertainty (K = 410, Estimate = 0.1[0.01 - 0.19], $p = 0.03$; test for heterogeneity: Q (df = 409) = 3000.22, $p < 0.0001$). At the order scale, 58 out of 86 (65%) orders showed neutral age-range size relationships, 27 were positive, and one was negative (Fig. 3C, and Supplementary Data 1). At the family scale, 310 out of 410 families (76%) showed neutral age-range size relationships, 84 were positive, and 15 were negative (Fig. 3D, and Supplementary Data 1). These results indicate that positive age-range size relationships are generally widespread regardless of the taxonomic scale, but detecting a significant relationship may be challenging at more narrow taxonomic levels (e.g., individual families), and effect sizes (Z-scores) differ substantially between families.

## Insularity constrains range size and modulates its relationship with species age

To test whether the age-range size relationship differs between species exclusive to islands (endemics) and those living on both island and mainland (or only on mainland), we ran a linear mixed-effect model. We found that species restricted to islands attained, on average, smaller ranges than species not restricted to islands (Estimate = −0.47, SE = 0.007, $t(25,240) = −59,26$, $p < 0.001$; Fig. 4, and Supplementary Fig. 4, Supplementary Table 5). However, the relationship between age and range size varied between species living exclusively on islands and species also occurring on the mainland. In general, species that were

restricted to islands exhibited a stronger positive effect of age on range size (Interaction term; Estimate = 0.15, SE = 0.01, $t(25,340) = 10.80$, $p < 0.001$; Fig. 4A) than species not restricted to islands (Estimate = 0.13, SE = 0.01, $t(24,300) = 10.80$, $p < 0.001$; Fig. 4B; and Supplementary Table 5). This pattern varied between taxonomic groups. The positive interaction term (larger ranges given age on islands) was supported for birds, reef fishes, and terrestrial mammals (Figs. 4D, F, H), whereas no interaction term was supported for amphibians or reptiles (Fig. 4C, E), and a negative term (smaller ranges given age on islands) for palms (Fig. 4G) (Supplementary Table 5). Hence, older species on islands generally have larger ranges, and younger species on islands have smaller ranges than expected based on age only. This may be due to island-specific biological and geographical processes (e.g., island ontogeny, the filling of 'empty' niches, diversification) that modulate range expansion.

## Dispersal modulates the age-range size relationship

To test whether the effect of age on range size varied with species dispersal abilities, we ran individual linear mixed-effects models for all species combined and for each broad taxonomic group. We used different dispersal-related traits to approximate dispersal ability for each group. Dispersal-related traits are an organism's attributes associated with movement, endurance, or colonization that may enhance dispersal[24,25]. As dispersal-related traits, we selected body size (for mammals, reef fish, amphibians and reptiles), egg type (for reef fishes), flight ability (for mammals), hand-wing index (for birds) and fruit size (for palms, see "Methods"). We included dispersal-related traits and age as interacting and fixed effects in the models. We included family, region (continents for terrestrial species: Americas, Asia, Africa, Europe, Australia, or marine regions for reef fish: Greater Caribbean, Tropical Eastern Pacific) and insularity (restricted to islands or not) as random effects.

We found positive correlations between age, dispersal ability and range size for all species combined, and for all taxonomic groups except for marine mammals. For marine mammals, body size correlated positively with range size, but age did not (Fig. 5, and Supplementary Table 6). Although the relationship between dispersal and range size was similar in strength to the relationship between species age and range size when all species were combined, dispersal ability showed a stronger effect for each taxonomic group, except for

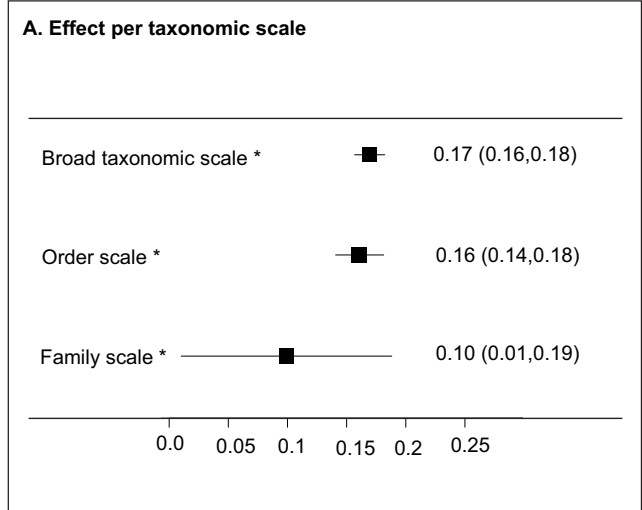

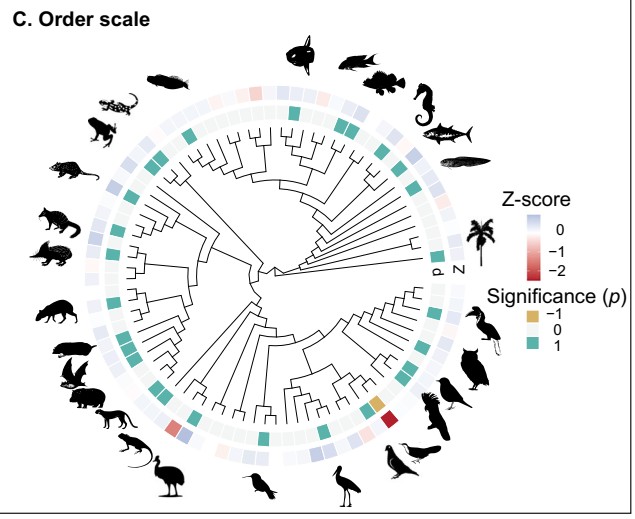

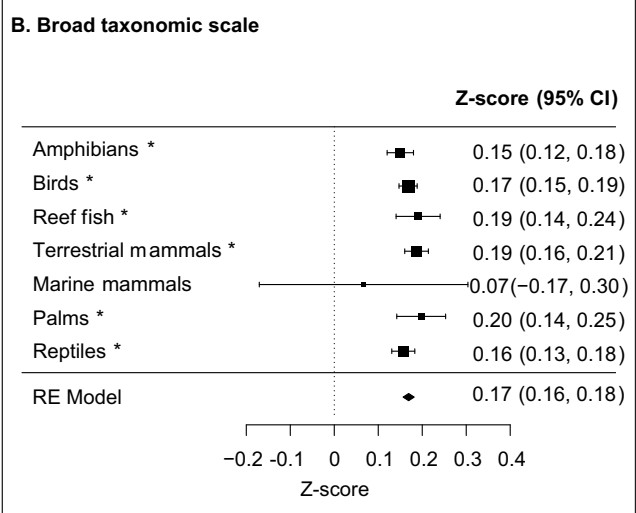

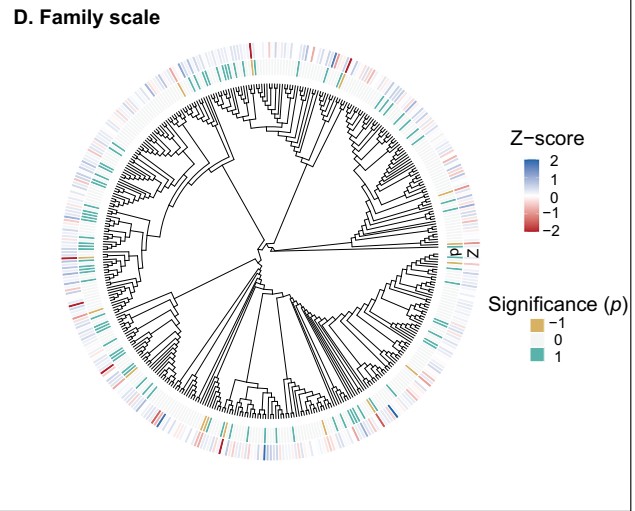

**Fig. 3 | Relationship between species age and range size across taxonomic scales. A** Overall relationship from a random effects (RE) model at the three taxonomic scales. **B** Forest plot of the species age-range size relationship for seven broad taxonomic groups, including the overall effect from the RE model. **C** Relationship at the order scale, where each tip in the phylogeny represents an order. **D** Relationship at the family scale, where each tip represents a family. We reconstructed the phylogenetic trees in TimeTree[94] by choosing a representative species for each order or family (Supplementary Data 2). In the circular phylogenies, outer arc colours indicate the effect size (Z-score) and inner arc colours represent significance (*p* < 0.05). Icons and asterisks indicate significant relationships for broad taxonomic groups and orders (see Supplementary Fig. 4 for an extended figure). *CI* Confidence Interval. See detailed information in Supplementary Data 1 for background data. For display purposes, seven Z-scores below −2 or above 2 at the family scale were set to (·) 2. All statistical tests were two-sided, and *p*-values were not adjusted for multiple comparisons, as the random effect model accounted for heterogeneity within the meta-analysis. Source data is provided as a Source Data file.

terrestrial mammals (Fig. 5, and Supplementary Table 6). In addition, we found that the strength of the relationship between age and range size depended on the species' dispersal abilities for all groups combined and for amphibians and reef fishes, but not for the other taxonomic groups (Interaction term; Fig. 5, and Supplementary Table 6). This result indicates that species with higher dispersal abilities (e.g., larger body sizes) have larger range sizes than expected based on age alone, probably because large-bodied amphibians and reef fishes rapidly attain maximum range sizes regardless of age, thereby reducing the positive effect strength of age on range size.

**Number of generations as a proxy of species age**

We evaluated whether species age accurately approximates the ecological mechanism behind range expansion from one generation to the next, by testing whether species' age expressed as the number of generations since speciation is a better predictor of range size than

species' age expressed as the number of years (Myr) since speciation. This analysis was only possible for birds and mammals due to limitations in generation time data, which were used to recalculate the number of generations based on species age expressed in years. We found that for both birds and mammals, the relationship between species age and range size is virtually the same when using the number of generations or the number of years (Figs. 5 and 6, and Supplementary Table 7). However, for terrestrial mammals, we detected an interaction term between number of generations and dispersal ability (Fig. 6), which was absent in the model with number of years (Fig. 5). Specifically, it shows that dispersal modulates the age-range size relationship, where the effect of species age on range size decreases with increasing body size. Furthermore, we found that species with aerial dispersal (bats) had larger ranges than terrestrial mammal species without flight abilities (Fig. 6, and Supplementary Table 7).

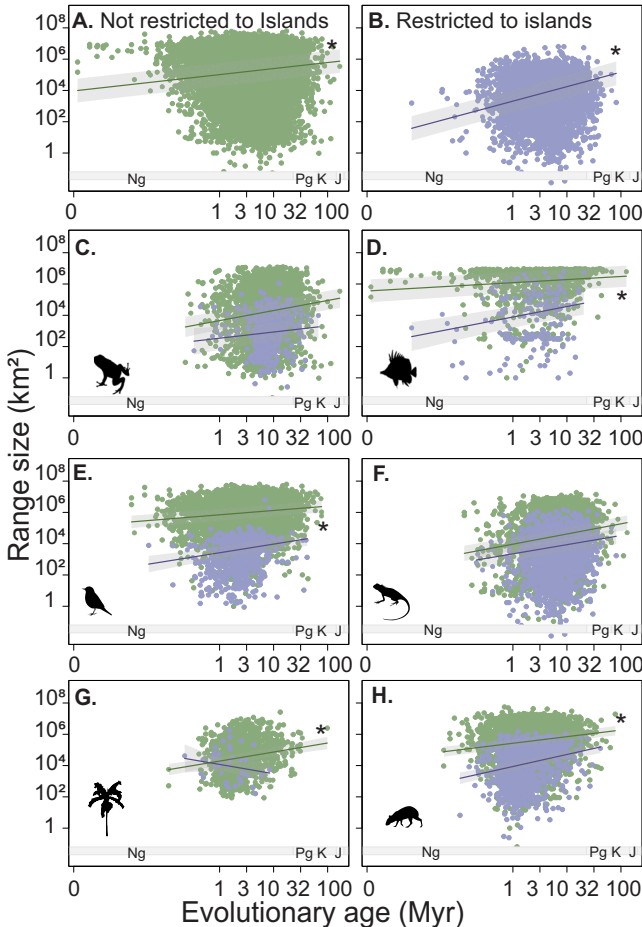

**Fig. 4 | Insularity modulates the relationship between age and range size.**
Relationship between age and range size for (**A**) all mainland species (i.e., species not restricted to islands) ($n = 22,255$), **B** all species restricted to islands ($n = 4161$), and for six broad taxonomic groups: **C** amphibians ($n = 4190$ species), **D**) birds ($n = 8785$ species), **E** reptiles ($n = 4837$ species), **F** reef fishes ($n = 1479$ species), **G** palms ($n = 1199$ species), and H) terrestrial mammals ($n = 5141$ species). We excluded marine mammals because insularity could not be measured. The solid line represents the predicted range (mean), while the shaded area denotes the 95% confidence interval (CI) around the prediction. *Myr* million years. *Asterisks* (*) denote a significant age-island interaction. Green = species not restricted to islands. Purple = species restricted to islands. Source data is provided as a Source Data file.

## Discussion

By analyzing data for 26,345 species from seven major taxonomic groups, we show that species age is positively associated with range size (except for marine mammals), that there is high variability among and within taxonomic scales, and that insularity and dispersal modulate the age-range size relationship. Older species with higher dispersal abilities that are not restricted to islands have larger range sizes than their younger counterparts with lower dispersal abilities and/or that are restricted to islands. The correlation between species age and range size is significant and, although modest, we demonstrate through null models that this relationship is unlikely to be due to chance. Furthermore, various sensitivity analyses, including those that adjust species age, exclude outliers, or use generation time instead of years as a proxy for species age, confirm the robustness of the species age-range size relationship. We show that evolutionary time for range expansion, the geographical context, and the species dispersal abilities collectively influence current species range sizes in both marine and terrestrial taxa.

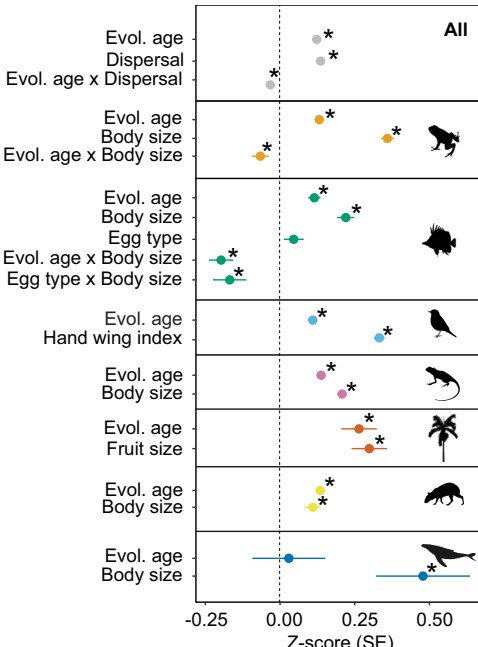

**Fig. 5 | Dispersal ability modulates the relationship between age and range size.**
Effect of age and dispersal-related traits on species range size for all species ($n = 25,355$), amphibians ($n = 4190$ species), reef fishes ($n = 1479$ species), birds ($n = 8785$ species), reptiles ($n = 4827$ species), palms ($n = 1115$ species), terrestrial ($n = 4919$ species) and marine ($n = 70$ species) mammals. When including all species (A), we only considered rescaled, continuous dispersal-related traits (e.g., body and fruit size). We considered 'flight ability', 'egg type', 'body size', 'hand-wing index' and 'fruit size' dispersal-related traits. Asterisks (*) denote significant relationships. Evol. age = evolutionary age; 'x' indicates an interactive effect. Points indicate the effect size (Z-score), with error bars showing the standard error (SE) for the Z-scores derived from linear mixed-effects models. In some cases, error bars are not visible due to small standard error (SE) values. Source data is provided as a Source Data file.

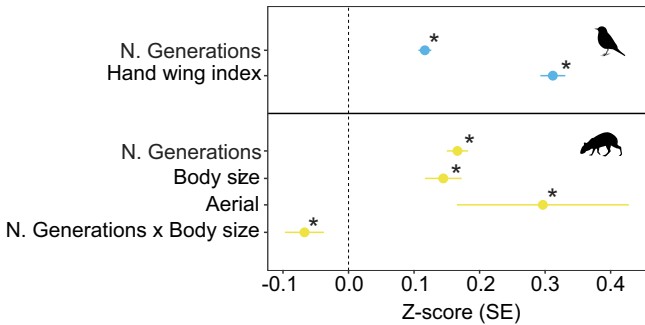

**Fig. 6 | Species age-range size relationships when number of generations is used as a proxy for age.** Relationship between the number of generations, dispersal-related traits and species range size for birds ($n = 7622$ species) and terrestrial mammals ($n = 4725$ species). We considered 'aerial dispersal', 'body size', and 'hand-wing index' dispersal-related traits for each taxon model. Asterisks (*) denote significant relationships. N. Generations = number of generations; 'x' indicates an interactive effect. Points indicate the effect size (Z-score), with error bars showing the standard error (SE) for the Z-scores derived from linear mixed-effects models. Source data is provided as a Source Data file.

Our meta-analysis demonstrates that while the positive effect of species age on range size is detectable across scales, it weakens at finer taxonomic levels. The mixed age-range size relationships (positive, neutral, or negative; Supplementary Table 1) reported in previous

studies may arise from analyses focused on a limited number of species at narrow taxonomic levels (e.g., within a single genus or family)[10] or restricted to specific geographical contexts, such as islands[25,26] or particular biomes, including the South African fynbos[27] and the Brazilian Atlantic forest[28]. Previous meta-analyses have shown that dispersal traits better explain range size variation at the family level than at broader taxonomic scales[22], emphasizing the role of species-specific characteristics in shaping distributions. Our findings align with these studies, suggesting that while species age itself may be a predictor of range size at broad taxonomic levels, likely reflecting the accumulation of dispersal events and geographic expansion over time, at lower taxonomic levels, range size may be more influenced by species-specific traits, such as dispersal ability or ecological specialization, which modulate the effect of age on range size[29]. These results show the importance of taxonomic scale in detecting macroecological patterns: broad-scale analyses can identify general correlates of species range size, such as age, whereas finer-scale analyses may not be able to detect such a relationship due to the increasing importance of additional factors that influence range size at lower taxonomic levels.

Insularity directly influences range size by imposing geographic constraints on the maximum range species can achieve and indirectly by altering the relationship between range size and species age. We show that island-restricted species have smaller range sizes compared to species that are not restricted to islands, consistent with previous findings on insular biotas[30,31]. Contrary to our expectation, the relationship between species age and range size is stronger for island endemics than for species that occur both on islands and on the mainland. The range size differences between young and old species are greater on islands than on the mainland, likely due to island dynamics and ontogeny shaping ecological and evolutionary processes. Without strong competition and predation pressures, early island colonizers may have experienced ecological release and niche expansion, consistent with supertramp strategies, in some cases, enabling them to achieve broader ranges[32–34]. On the other hand, the younger, narrow-ranged island endemics are likely a result of higher speciation rates and adaptive radiations driven by intense competition and specialization[35]. Highly diversified lineages on islands, such as in the vascular flora of the Canary Islands, tend to have a much higher proportion of small-ranged species, whereas less diverse lineages contain more widespread species[36]. The balance between ecological release and adaptive radiation may vary depending on island connectivity and age. More isolated or younger islands tend to foster ecological release, while older or more connected islands may be hotspots for adaptive radiations, as exemplified by the Galápagos finches and Hawaiian *Drosophila*[37]. Island species tend to be younger (Supplementary Table 8, Supplementary Fig. 5), which is consistent with theoretical models on island speciation[38]. Moreover, island size might play a role because species colonizing larger islands are less likely to go extinct[33], allowing them to attain large ranges and reach older ages. Finally, many older species restricted to islands may represent relics of once more widespread lineages that have suffered extinction in other parts of their former range, as observed in some island bird species[39]. While most taxa conform to the age-range size pattern, we found an exception among island-restricted palms. Contrary to expectations, older palm species on islands do not exhibit larger ranges. This anomaly may be attributed to anthropogenic factors, such as habitat transformation, that have led to range size reductions of evolutionary old (and possibly phylogenetically unique— i.e., descending from long branches) palms. Our findings suggest that the stronger positive relationship between species age and range size on islands may reflect the interaction between island age, isolation, and ontogeny.

Dispersal was associated with range size across all taxa, as shown in a previous meta-analysis[22]. Based on this, we hypothesized that the correlation between species age and range size would be weaker or absent in species with higher dispersal abilities, because high dispersal can accelerate the range expansion process, enabling species to obtain large ranges more rapidly than less dispersive species[21]. Indeed, less dispersive reef fishes and amphibians show a stronger correlation between species age and range size, and terrestrial mammals when age is measured as the number of generations instead of chronological years. Because dispersal is often an intergenerational event, typically occurring during the propagule or juvenile stages to avoid kin competition or during the final phase when individuals seek new territories before or after reproduction[40], the number of generations may be a more relevant descriptor for the demographic processes underlying range evolution. Although age showed a weaker effect on range size than dispersal, its additive effect supports the notion that time for range expansion and dispersal ability are both crucial for range size[17,22,41]. The lack of support for a more pronounced positive relationship between age and range size in less dispersive species of reptiles, birds, palms and marine mammals may be attributed to historical and biological differences among these groups. For instance, palms with large fruits often rely on large-bodied fruit-eating and seed-dispersing animals (frugivores) for seed dispersal and range expansion - the coconut being an exception, as it currently relies on water dispersal[42]. As large-bodied frugivores move seeds over long distances in their often large home ranges[43], palms with large fruits may expand their range faster than palms with small fruits. However, this fast-range expansion of large-fruited palms did not obscure the effect of age on range size. Possibly, the Quaternary extinctions and ongoing declines of many large-bodied animals (megafauna) have led to dispersal limitation of palm species with large fruits, leading to local extinctions and range reductions[44,45], and the recovery of a positive age-range size relationship. Large body sizes in marine mammals come with energetic constraints, such as the need to forage over extensive areas and greater fasting capacities, enabling more dispersive species to attain wider global distributions[46]. Thus, the lack of an age-range size relationship in marine mammals might result from their high dispersal capacity. Another possible explanation for the absence of an age-range size relationship in marine mammals is the reduced statistical power to detect significant patterns, as indicated by the error bars in Fig. 3. A sensitivity analysis using random samples from the full dataset (Supplementary Fig. 6) suggests that positive effects are harder to detect and more variable when sample sizes are small. This limitation makes interpreting the results more challenging for marine mammals or any study using few species. While traits that confer high dispersal abilities, such as large body size, may allow species to attain larger ranges, the specific dispersal trait relationship to range size may be more complex and clade-dependent. The selection of relevant and meaningful dispersal traits is crucial, as it can significantly influence the effect sizes observed in the pattern of interest. Sometimes, a combination of traits better reflects the overall dispersal capacity of an organism[22].

Several factors beyond those we examined here may limit the strength of the relationship between age and range size. First, historical events such as paleoenvironmental changes, tectonics, and vicariance have shaped present-day biodiversity and distribution patterns by driving differential speciation, extinction and range dynamics, and species-driven responses such as habitat shifts[47–49]. For example, the last glacial period pushed species to lower latitudes, and/or confined them to smaller local glacial refugia[50], while vicariance events often result in asymmetrically split ranges[51]. Second, human-induced range contractions and extinction, such as the ones occurring during the Quaternary but also nowadays, have particularly affected ranges of large-bodied animals and indirectly range sizes of plants that depend on them for dispersal[52–55]. The detection of recently diversified or near-extinct species and the accurate estimation of range size, is impacted by unnatural (e.g., human-driven) extinction and extirpation events, and geographic sampling biases[13,47,56]. Third, taxonomic classification biases, stemming from lumping or splitting methods, can impact

species-level phylogenies and hence age and range size estimates[57–59]. Moreover, taxonomic classification biases are latitudinally dependent, where underestimating the number of species in the tropics might result in overestimating range and age[60]. Fourth, species age estimates from phylogenies can be overestimated because they ignore extinction and anagenesis[13]. For instance, the death of a sister species can artificially inflate the age of the surviving species[38]. Adjusting for extinction from constant birth-death model estimates does not lead to substantially different age estimates (congruence between 73–96%, depending on the extinction fraction[23]), probably because estimating extinction rates from phylogenies is challenging and may bias age estimates by itself. We show that using an adjusted species age did not change our conclusions, likely because this age adjustment primarily influences longer branches (i.e., older species). Older species with longer branches have higher probabilities of extinction or speciation, and their ages reach an asymptote after a certain point. Hence, those species with the oldest non-adjusted age (i.e., the ones with the longest branch length) still remain among the oldest when age is adjusted, but with less extreme values. Furthermore, the largest error in estimating species ages from phylogenetic trees is linked to an incomplete sampling of extant species[23], and we minimized this risk by only including phylogenies with comprehensive species representation. Although there is much debate on the adequate way of measuring species age[13], the concept of age as we used it here, defined by the time to the most recent common ancestor, is currently the only aspect that can be practically measured through existing molecular phylogenetic methods[38].

Our study investigated how species' evolutionary age, along with its interaction with insularity and dispersal, shapes species range size. We show that species with smaller ranges, which are often linked to heightened extinction risk, are disproportionately restricted to islands, tend to have lower dispersal capacities, and frequently evolved recently, consistent with the ephemeral speciation model, in which species may quickly arise but die young[61]. This limited range size is also a critical factor explaining why island endemic species are disproportionately endangered or have recently gone extinct[30]. The evolutionary loss of dispersal abilities, a phenomenon commonly observed in insular species, further exacerbates their extinction risk by constraining their ability to expand their ranges and respond to environmental changes[36,62]. Islands, while often biodiversity hotspots, can act as eco-evolutionary "traps," where the combination of restricted ranges, physical barriers to dispersal, evolutionary loss of dispersal traits, and local extinctions intensifies the risk of lineage extinction. While older species typically have larger ranges, many exceptions exist, such as "living fossils" that are species threatened by extinction because of being confined to small ranges despite their evolutionary longevity[63]. By integrating geographical context, clade-specific traits, and phylogenetic history (species age), we provide a more nuanced framework for assessing extinction risk. There is a need for targeted conservation efforts, particularly for island ecosystems and range-restricted species disproportionately impacted by natural and human-driven threats. Ultimately, our study sheds light on how ecological, evolutionary, and geographical processes might shape species' vulnerability to extinction simultaneously. Understanding these dynamics is critical for predicting future biodiversity loss and informing conservation strategies that safeguard the persistence of Earth's species in an era of unprecedented environmental change.

## Methods
### Data collection
**Range size.** We compiled range size data for 27,145 species (Supplementary Fig. 1), using distribution maps from the IUCN Red List (IUCN Version 6.2 2007)[64] for amphibians, reptiles and both marine and terrestrial mammals. For birds, we used data from the AVONET database[65]; for palms, from Hill et al.[66]; and for reef-associated bony

fishes (class 'Actinopterygii'), from Robertson & van Tassell[67] and Robertson & Allen[68]. We documented and/or approximated range size as the extent of occurrence (EOO), using equal-area projections (e.g., Mollweide or WGS84). We chose EOO because it reflects potential range expansion over time while minimizing bias in estimates of natural (i.e., pre-human) range sizes due to local extinction, habitat fragmentation, and other factors (e.g., ecological niche) that influence range size independent from the expansion process.

For 5575 terrestrial and 70 marine mammal species, we calculated range size using the 'st_area' function from the R package 'sf'[69] using the Mollweide projection. Based on the IUCN data, only 'extant' and 'native' species were included. For marine mammals we only considered "exclusively marine", excluding three species that "occur on mainland". For 8,785 bird species, range size was measured as the total area of the mapped range from BirdLife International and was restricted to areas of the range coded as Presence = 1 (extant only), origin = 1 & 2 (native and reintroduced), and seasonal = 1 & 2 (resident and breeding). The total combined mapped area was originally calculated using the 'areaPolygon' function from the R package 'geosphere'[70], which calculates the area in the World Geodetic System (WGS84) projection using spherical distances. We restricted our analysis to species in terrestrial habitats, thus excluding marine birds. For 4190 amphibian species, we calculated range size using the 'area' function from the R package 'raster'[71] with the Mollweide projection. For 5844 reptile species, we calculated range size using the 'st_area' function from the R package 'sf'[69] and the Mollweide projection. We excluded 'extinct' and 'extinct in the wild' species. For 1481 reef-associated bony fish species (class 'Actinopterygii'), we measured range size as the extent of occurrence (convex hull area) using the function 'CalcRangeSize' from the package 'speciesgeocodeR'[72]. We excluded records occurring on land using the function 'cc_sea' from the R package 'CoordinateCleaner'[73]. For 1200 palm species with range size data reported in ref. 66, we excluded species with range size values of 0.01 km², because those were assigned an arbitrarily small number due to deficient data (less than 3 records).

**Evolutionary age.** We estimated evolutionary age for 26,346 species as the median branch length of the terminal nodes (i.e., species or tips) across 100 phylogenetic trees (Supplementary Fig. 1). We obtained 100 phylogenetic trees for birds[74], squamates[75], amphibians[76] and mammals[77] from 'Vertlife.org', for fish from the 'Fish Tree of Life'[78], and for palms from[79]. The full phylogenies included 31,516 tips for fish, 9993 tips for birds, 9755 tips for reptiles (Squamata), 2539 tips for palms, 7239 tips for amphibians, and 5911 tips for mammals. To mitigate overestimating species age due to sampling bias, we estimated species ages using the complete phylogenies, then retained only species for which range size information was available. This yielded species age estimates for 1479 fish, 1199 palms, 5482 reptiles, 4190 amphibians, 8785 birds, 70 marine mammals and 5141 terrestrial mammals.

**Dispersal.** We collated data on species dispersal abilities for 26,474 species. For 5063 mammal species, we obtained data on body size (continuous: body mass in g) and flight ability (categorical: yes = 1, no = 0) from Phylacine 1.2.1[80,81]. For 8785 bird species, we obtained hand-wing index (continuous: HWI) data from AVONET[65]. For 4190 amphibian species, we obtained body size data (continuous: snout-vent length in Anura, and total length in Gymnophiona and Caudata in mm) from AmphiBIO[82] and for 5,830 reptile species (continuous: maximum length in mm) from Feldman et al.[83] and Meiri[84]. For 1481 reef fish species, we obtained body size data (continuous: maximum length in cm) and egg type (categorical: pelagic = 1, non-pelagic = 0) from Alzate et al.[85], Robertson & van Tassell[67] and Robertson & Allen[68]. For 1116 palm species, we obtained fruit size (average fruit length in mm) data from Kissling et al.[86].

**Insularity.** We collated data on insularity (land masses smaller than Australia) for 26,784 species. We classified species as "restricted to islands" if found exclusively on islands. Conversely, species "not restricted to islands" are those found on continents or on both continents and islands. Despite some island endemic species inhabiting multiple islands, their maximum potential range is still generally less than that of continental species or species that live both on continents and islands. For 5285 terrestrial mammal species, we obtained data on island endemicity from Phylacine 1.2.1[80,81]. We considered terrestrial mammals not restricted to islands as those living on 'mainland'. For 8785 bird species, we collated data on insularity from Sheard et al.[87]. We considered bird species as "restricted to islands" when they were reported to be 100% associated with islands. For 4190 amphibian species, we obtained information on insularity by overlaying species distribution maps with a shapefile of islands of the world. We generated a shapefile layer delimiting all islands from the world based on a digital elevation model at 30 m pixel resolution. Using this shapefile we conducted a spatial join with the IUCN amphibians ranges, categorizing amphibian species as 'restricted to islands' if only they were located in islands. For 5844 reptile species, we obtained information on insularity ("restricted to islands" vs "not restricted to islands") from Meiri[84]. For 1481 reef fish species, we obtain data on insularity (occurring only on islands, and occurring on the continent or both continent-island) from Alzate et al. [85], Robertson & van Tassell[67] and Robertson & Allen[68]. For palms, we classified 1199 species as restricted or not restricted to islands following Cassia-Silva et al.[88]. We did not consider marine mammals as restricted to islands as most have widespread distributions.

**Region.** We assigned terrestrial mammals, reptiles, amphibians and bird species to continents (Americas, Africa, Asia, Australia, Europe) and reef fishes to marine regions (Greater Caribbean, Tropical Eastern Pacific) based on the centroid from the distribution maps. We did not assign marine mammals to a specific region as most have widespread distributions.

**Number of generations.** We estimated the number of generations since the species' origin by dividing the species' age by generation time. We compiled generation time data for 4725 mammal species from Pacifici et al. [89] and for 7499 bird species from Andermann et al.[90]. For simplicity, we assume non-overlapping generations (i.e., each generation completely replaces the previous one).

## Data analyses

To examine the overall effect of age on range size, we ran a linear mixed model including 26,346 species from seven broad taxonomic groups (amphibians, reef fishes, birds, reptiles, palms, terrestrial and marine mammals). We used the function 'lmer' from the 'lme4' R package[91]. To account for phylogenetic structure, we included 'family' nested within 'order' nested within 'broad taxonomic group' as a random effect. We also ran individual models for each taxonomic group with the same random effect structure (family nested within order), except for reptiles for which only family was included, and palms for which no random effects were included.

We tested whether the effect of species age on range size is expected by chance by building null models in which we randomized the median species' age 1000 times. We ran these null models for the complete set of species and each broad taxonomic group. We built a linear mixed-effects model for each randomisation. The random effect structure was the same as for the main model. We considered the effect of age on range size to be expected by chance if the observed effect size falls within the distribution of the 1000 effect size from the null models.

To examine how the age-range size relationship varied among taxonomic groups, we ran linear models at three taxonomic levels: at

the broad taxonomic level (reef fishes, birds, terrestrial mammals, marine mammals, amphibians, squamates and palms), at the order level (88 orders), and the family level (418 families). We performed a meta-analysis for each taxonomic level to test the overall effect of age on range size based on all individual relationships. We fitted a random effects model, including the Z-score values with their corresponding standard errors, using the function 'rma' from the R package 'metafor'[92].

We tested the effect of insularity on the relationship between age and range size by running a linear mixed model with species age and insularity as additive and interacting fixed effects and 'family' nested within 'order' nested within 'broad taxonomic group', and continents (Americas, Africa, Asia, Australia, Europe) or marine regions for reef fishes (Greater Caribbean, Tropical Eastern Pacific) as random effects. We also ran individual models for each taxonomic group, but with different random effect structures. We included continent, and 'family' nested within 'order' for all groups, except for reptiles for which we included continent and 'family' as random effects, and palms for which we only included continent as random effect. Marine mammals were excluded from the analysis as too few species were restricted to islands. We used the function 'lmer' from the 'lme4' R package[91].

To test whether the relationship between age and range size depended on dispersal, we ran linear mixed-effects models for the six broad taxonomic groups (amphibians, reef fishes, birds, reptiles, palms, terrestrial and marine mammals). We included age and dispersal-related traits as additive and interactive fixed effects. We accounted for insularity and palaeogeographic history by including whether species are restricted to islands (restricted or not restricted to islands) and continents (Americas, Africa, Asia, Australia, Europe) or marine regions for reef fishes (Greater Caribbean, Tropical Eastern Pacific), as random intercepts, except for marine mammals, which are all restricted to the open ocean and have global, circumtropical or circumtemperate distributions. To account for phylogenetic structure, we included 'family' nested within 'order' as random slopes for birds, terrestrial and marine mammals, reef fishes and amphibians. For reptiles, we only included 'family' as a random effect, and for palms we did not include taxonomic structure as a random effect.

To test whether the number of generations is a better proxy of species age than the number of years, we investigated whether using different species' age metrics (millions of years vs. number of generations) influences the relationship between species age and range size. To this end, we ran linear mixed models including fixed factors: species age, dispersal traits (Hand-wing Index for birds and aerial dispersal and body size for mammals) and random effects: 'family' nested within 'order', Island, and Region.

To meet linearity assumptions, dispersal-related traits, age and range size were $log_{10}$-transformed and dispersal-related traits were rescaled using the function 'rescale' from the 'arm' R package[93]. All models were standardized using the function 'standardize' from the 'arm' R package[93].

## Statistics & reproducibility

No statistical method was used to predetermine sample size. Instead, sample sizes were determined by the availability of data for each taxonomic group. No data were excluded from the analyses unless they lacked key variables (e.g., missing range size, age estimates, insularity or dispersal data) or had implausible values (e.g., range size = 0). All inclusion criteria and data cleaning steps are detailed in the "Methods" section.

## Reporting summary

Further information on research design is available in the Nature Portfolio Reporting Summary linked to this article.

## Data availability

The datasets generated during the current study are available in fig-share (https://doi.org/10.6084/m9.figshare.25134749). Source data are provided with this paper.

## Code availability

The code and data necessary for reproducibility are available in Zenodo (https://doi.org/10.5281/zenodo.15316357).

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

## Acknowledgements

This study was funded by the German Centre for Integrative Biodiversity Research (iDiv) Halle-Jena-Leipzig through a German Research

Foundation grant (DFG FZT 118, 202548816; AA, RR, REO). Additional support was provided by sDiv, the synthesis center of iDiv with an individual postdoctoral grant (DFG FZT 118, 202548816; AA, RR), a Marie Skłodowska-Curie postdoctoral grant funded by the European Union's Horizon 2020 research and innovation programme (Grant No. 101061723; AA), the Swedish Research Council (2017-04980, 2022-03927; CDB), the University of Gothenburg Strategic Research Area in Biodiversity and Ecosystem Services in a Changing Climate (BECC; CBD), a DGAPA/PAPIIT UNAM grant (IA206523; JAV), the Swiss National Science Foundation (grant No. 310030_188550; LP). We thank all data collectors and providers.

## Author contributions

Conceptualization: A.A., J.R., L.P., R.R., R.E.O.; Methodology: A.A., T.J., J.A.V., A.Z., Fvd.P., R.E.O.; Formal analysis: A.A.; Data Curation: J.A.V., A.H., C.D.B., J.T., D.R.R., A.A.; Visualization: A.A.; Writing–Original Draft: A.A.; Writing– Review & Editing: A.A., A.Z., R.R., Fvd.P., J.A.V., D.R.R., A.H., C.D.B., J.T., D.R.R., T.J., L.P., J.R., R.E.O.

## Funding

## Competing interests

The authors declare no competing interests.
