## [Peer Review file · Nature Communications]

Evolutionary age correlates with range size across plants and animals.

Corresponding Author: Dr Adriana Alzate

Version 0:

Reviewer comments:

Reviewer #1

(Remarks to the Author)

This is an interesting manuscript, well-written, based on a massive and impressive dataset, dealing with age-area relationships, a very interesting issue from a conceptual and theoretical point of view and important for better establishing effective conservation actions. Despite this importance, this seems to be (to me) a difficult problem to deal with and “resisted” nice solutions for a while, mainly because it is hard to get nice data for testing the relationship, which is actually my main concern with results and conclusions. In short, there are two main issues about the estimate of species’ age and the relationships with area.

About the first issue, species’ age, authors did a nice discussion of the main pitfalls and assumptions starting from line 288, but the problem is exactly that these are not solved, and I think that they strongly jeopardize the conclusions. I think that estimating species’ age by the age of each tip to the MRCA, as used by authors, does not capture the theoretical dynamics involved in the age-area relationships. Part of the problems are pointed out in the discussion started in line 288, but there are other issues. For instance, authors point out that “correcting for extinction from constant birth-death model estimates does not lead to substantially different age estimates”. Indeed, this is nearly obvious because when using age to MRCA ASSUMES a birth-death model, and the point is that it is not obvious that this assumption holds. The reason is exactly because the extinction component of diversification is not the only process affected by other life-history and ecological traits, and the speciation dynamics is also affected by these traits and may interact. Thus, the length from tips to MRCA does not necessary reflect species’ age in a demographic sense (that matches the ecological dynamics involved in the age-range relationship). From a practical/operational point of view, for instance, the age is measured in millions of years from MRCA to tips, and ecological and evolutionary dynamics occurs in generation time (which vary a lot among the species; authors discuss in the previous paragraph how body size relates to dispersion ability, and thus affect range size, but it is also related to age due to generation length component). The absence of age-area relationships due to geographic constraints for islands is obvious and intuitive (which is related to the response variable, range size, and authors discuss other problems there), but the other interpretations of the variables and their relationship may be more challenging. Finally, although authors discuss the taxonomic issues, it would be nice to be more explicit about the latitudinal taxonomic gradients more recently discussed by Freeman and Pennell (2021, <https://doi.org/10.1016/j.tree.2021.05.003>).

The above problems are quite resistant to solutions for broad-scale, interspecific studies as proposed in this paper. One of the assumptions is that all these confounding factors at the macroscale may be random errors and increase the noise in data, but even so, the main trends would be potentially detectable. In this sense, it is quite common that these macroecological relationships are better viewed as constraint envelopes rather than overall functional (linear or exponential) relationships. Moreover, the above issues most likely produce bias whose directions are hardly predictable. Indeed, as can be seen in figures 1 and 3 of the manuscript, the effect size is TOO low.

Following the issues raised about the low effect size revealed in Figures 1 and 3, it is important that, although there may be statistical significance effects, there are potentially inflated Type I errors due to phylogenetic autocorrelation, and thus, I’m afraid that most of the coefficients would be statistically closer to zero than shown. It is unclear to me that phylogenetic structure was accounted for in the mixed model, and thus, what is the meaning of the P-values shown throughout the manuscript.

I’m not saying that all the above issues would completely jeopardize the manuscript’s conclusions, and I applaud the authors’ efforts to attack this important problem. At the present time, I’m a bit skeptical about the result, but if other reviewers

and authors are confident about their findings, I have no hard feelings about this.

Reviewer #2

(Remarks to the Author)

This manuscript demonstrates the positive relationship between evolutionary age of species and their range size. The conclusions drawn from the results seem valid and I detect no serious methodological issues. The relationships are not strong, but this is discussed, and they are based on a lot of data, across many species. Overall, I think this manuscript is an interesting and useful contribution. I have a few suggestions for clarifications.

Suggested improvements and points to address:

1. The title is a bit complex.
2. Lines 56-59: Please provide references that directly address these important arguments or convey their uncertainty. These arguments are repeated throughout the manuscript and therefore need a good foundation.
3. Not all island species are restricted to one island, their range size can cover many islands. Additionally, some species occur both on islands and on mainland. Therefore island-dwelling is not necessarily equal to being restricted to a small range. Please discuss how this affects the assumptions and conclusions of this study.
4. Lines 91-101: Consider clarifying this long sentence by splitting it.
5. Lines 77-88 and 157-173: Using the broad description of "geographical constraints" seems inappropriate because not all types of geographical constraints (e.g. mountains) have been tested here and this study instead only refers to island vs. non-island species.
6. A clear definition of what is meant by a dispersive species is lacking in the introduction but is needed here already to understand the arguments.
7. Hypotheses 1 and 2 are very similar and are addressed together in the first results section. Other hypotheses have their own results sections. This can be confusing to the reader. Consider combining the first two hypotheses or splitting the first results section or another solution for clarification.
8. Lines 113-114: "These results were robust to outliers". It seems important to highlight this more (e.g. in figure caption) because the effect of outliers does come to mind when looking at the figures. The additional outlier test makes the results convincing.
9. Lines 164-169 and Figure 3: The use of dashed lines to indicate both significant and non-significant relationships is confusing.
10. Lines 216-320: There is no header for the discussion.
11. Lines 267-273: More nuance is needed in the discussion about the choice of proxies for dispersal ability. Large body mass or large fruit size is not always an indicator of high dispersal ability, small animals (birds) and small fruits (eaten by large animals or by bats that fly far) can also disperse over long distances.
12. Lines 306-309: "species with smaller ranges are more often threatened with extinction". This seems to be a new result presented in the discussion. I cannot find the description of this in the supplementary methods. It is therefore unclear to me what has been done here. Please be aware that many IUCN statuses have been assigned based directly on the range size (Gaston & Fuller 2009, Journal of applied ecology). Therefore, any tests of relationships between range size and IUCN status would not be useful.
13. Lines 347-356: Please clarify if this criterion is true for all groups: "We considered bird species as "restricted to islands" when they were reported to be 100% associated with islands". How were species categorized that occur both on mainland and islands?
14. Lines 357-359: Please indicate what this is used for. Is it to select species as described in lines 324-327?
15. Please provide more clarity earlier on in the manuscript about what the dispersal related traits are exactly.
16. A discussion is missing of the possible influence of a difference between present-day range size and range size before human disturbance. This seems crucial.

Version 1:

Reviewer comments:

Reviewer #1

(Remarks to the Author)

(Remarks on code availability)

.

Reviewer #2

(Remarks to the Author)

After carefully reading through the new manuscript and the responses to the reviewer comments, I have no further questions. I think all points have been addressed well, the methods are clearer and the discussion is now more comprehensive.

These are my only minor comments:

- Are the R code formulas used for the linear mixed-effects models written out somewhere (supplement)? It is a bit unclear otherwise how the nestedness levels were done technically.
- Small paragraph blocks need to be indicated in the text to make it easier to read.

(Remarks on code availability)

RESPONSE LETTER

We have now included additional sensitivity analyses to respond to several concerns from reviewer 1: 1) we used null models to test whether the effect of age on range size is not produced by chance. 2) we confirmed the robustness of the species age-range size relationship using a measure of range size that overcomes issues related to age overestimation due to unknown extinction-speciation events. 3) We tested whether using generation time, instead of years, as a proxy for species age would impact our results. 4) We tested what is the effect of sample size on detecting significant age-range size relationships.

Moreover, we have included two main changes in our manuscript. 1) We estimate the overall evolutionary age per species as the median branch length of the terminal nodes (i.e., species or tips) from 100 phylogenetic trees instead of the mean. The median is a more adequate statistic as it is less affected by extreme values. 2) We corrected the code used to merge species attributes with the phylogeny, which impacted some results related to insularity and dispersal. While the main conclusions remain unchanged, some specific results have been updated. For example, the relationship between dispersal and range size for certain groups and the relationship between age and range size for island-restricted species have been revised. In addition, we have incorporated all the recommendations and believe that these have improved our study. Our responses are indicated by “R/” and in blue.

Best Regards,

Adriana Alzate, on behalf of all co-authors

REVIEWER COMMENTS

Reviewer #1 (Remarks to the Author):

1. This is an interesting manuscript, well-written, based on a massive and impressive dataset, dealing with age-area relationships, a very interesting issue from a conceptual and theoretical point of view and important for better establishing effective conservation actions. Despite this importance, this seems to be (to me) a difficult problem to deal with and “resisted” nice solutions for a while, mainly because it is hard to get nice data for testing the relationship, which is actually my main concern with results and conclusions. In short, there are two main issues about the estimate of species’ age and the relationships with area.

R1. We appreciate the reviewer's interest in our study and thank them for highlighting the challenges of addressing this question. We refer to our responses to their concerns below.

2. About the first issue, species’ age, authors did a nice discussion of the main pitfalls and assumptions starting from line 288, but the problem is exactly that these are not solved, and I think that they strongly jeopardize the conclusions. I think that estimating species’ age by the age of each tip to the MRCA, as used by authors, does not capture the theoretical dynamics involved in the age-area relationships.

R2. We appreciate the concern that estimating species' age based on the tip-to-MRCA age doesn't fully capture the dynamics of the age-area relationship. We agree that the risk of simple methods to estimate the age of a species is that these overlook unknown extinction or speciation events, thereby leading to an overestimation of species age (Hodge, J. and Bellwood, D.R. 2015 <https://doi.org/10.1111/geb.12264>, Calderón del Cid et al. 2024 <https://doi.org/10.1111/geb.13890>). We have now addressed the issue of

overestimating species age due to unknown extinction or speciation events by implementing the alternative method proposed by Calderón del Cid et al. 2024. Their method uses an extant phylogenetic tree, applies a birth-death model, estimates the probability of extinction and speciation events, and yields both the original (based on estimated branch lengths) and adjusted species' ages (see more details in Supplementary Table 4, Supplementary Figure 2).

We tested how the choice of species age metric influences the outcomes of the relationship between species age and range size by running statistical models for all groups together and for each of the seven taxonomic groups (birds, terrestrial and marine mammals, amphibians, reptiles, reef fishes and palms) separately (see more details in Supplementary Table 4). The outcomes of all statistical models using the adjusted species age (i.e. using the Calderon del Cid et al (2024) method) show nearly identical results to models using only branch lengths (See figure below). We included these results in Supplementary Figure 2 and Supplementary Table 4. We refer also to this in lines 122-125 and 388-402.

Left: Relationships between species evolutionary age (median age across 100 phylogenetic trees) and range size. **Right:** Relationships between species-adjusted evolutionary age (median age across 100 phylogenetic trees) and range size. a) all species (N = 26,345 species), and b-h) separated for the seven taxonomic groups: b) amphibians (N = 4,190 species), c) reef fishes (N = 1,479 species), d) birds (N = 8,785 species), e) reptiles (N = 5,481 species), f) palms (N = 1,199 species), g) terrestrial mammals (N = 5,141 species), and h) marine mammals (N = 70 species). The black lines denote a significant relationship between evolutionary age and range size. Geological periods are denoted as Ng = Neogene, Pg = Paleogene, K = Cretaceous, and J = Jurassic. Data were log₁₀-transformed for analyses and plotting. N = number of species. Mya = million years ago.

Biases related to the lack of information on potential extinction events or topological uncertainty are unlikely to affect our study's conclusions. While our approach allows us to evaluate the potential effects of extinction events, there are challenging to estimate extinction using phylogenies based solely on extant taxa. Nevertheless, species' age significantly correlates with range size in all taxonomic groups (except for marine mammals), irrespective of how species age is quantified, based on 100 phylogenetic trees.

The lack of a change in the results using the adjusted species age is likely because this adjustment primarily influences longer branches (i.e., older species). Older species with longer branches have higher probabilities of extinction or speciation, and their ages reach an asymptote after a certain point. Hence, those species with the oldest age in our original manuscript (i.e. the ones with the longest branch length) still remain among the oldest with our new age quantification method, but with less extreme values. Here is an example for amphibians:

For comparability with the previous literature and interpretation, we kept branch length as a proxy for species age in the main text but included the outcomes of models using the adjusted age in supplementary materials. We mention this also in the discussion section lines 404-405.

3. Part of the problems are pointed out in the discussion started in line 288, but there are other issues. For instance, authors point out that “correcting for extinction from constant birth-death model estimates does not lead to substantially different age estimates”. Indeed, this is nearly obvious because when using age to MRCA ASSUMES a birth-death model, and the point is that it is not obvious that this assumption holds.

R3. Indeed, following Calderón del Cid et al. 2024 (<https://doi.org/10.1111/geb.13890>), replacement speciation, e.g. a Birth-Death process, has to be assumed to perform any way of correcting the species age for previous speciation/extinction events. It is almost certain that most empirical trees are not Birth-Death trees, and that extinction rates or speciation rates are not constant across the phylogeny. Unfortunately, extinction rates are difficult to estimate from molecular phylogenies with only extant taxa (see Rabosky 2010, <https://doi.org/10.1111/j.1558-5646.2009.00926.x>). We have included this in the discussion in 388-405.

4. The reason is exactly because the extinction component of diversification is not the only process affected by other life-history and ecological traits, and the speciation dynamics is also affected by these

traits and may interact. Thus, the length from tips to MRCA does not necessarily reflect species' age in a demographic sense (that matches the ecological dynamics involved in the age-range relationship). From a practical/operational point of view, for instance, the age is measured in millions of years from MRCA to tips, and ecological and evolutionary dynamics occurs in generation time (which vary a lot among the species; authors discuss in the previous paragraph how body size relates to dispersal ability, and thus affect range size, but it is also related to age due to generation length component).

R4. Thanks for raising this issue and the suggestion to focus on generation time to better approximate the ecological dynamics involved in the age-range relationship. To investigate whether generation time better reflects the age of species in a demographic sense, and is thus possibly a better metric to investigate the relationship between species age and range size, we collated data on generation times for mammals (4725 species, <https://doi.org/10.5061/dryad.gd0m3>) and birds (7499 species, <https://doi.org/10.1111/ecog.05110>). For other groups, this information is unfortunately not available.

First, we tested whether there is a correlation between generation time and species age using linear mixed models that included species age as a response variable, generation time as a fixed factor, and as random effects: family nested within order, insularity and continent. We did not find a correlation between generation time and species age for birds (Estimate = -0.056, SE = 0.037, df = 6889, t = 1.523, p = 0.128) or mammals (Estimate = -0.035, SE = 0.05, df = 4673.08, t = -0.695, p = 0.487), which suggests that species generation time does not have a direct influence on the current 'evolutionary' age of the species.

In addition, we calculated the number of generations (assuming non-overlapping generations) since the species' origin by dividing the species' age by generation time. We investigated whether using different species' age metrics (millions of years vs. number of generations) influences the relationship between species age and range size by running linear mixed models for birds and mammals, including fixed factors: species age, dispersal traits (Hand-wing Index for birds and aerial dispersal and body size for mammals) and random effects: family nested within order, insularity, and continent.

Results are virtually the same for birds and similar in mammals: direction and significance (See figures below). However, for terrestrial mammals, using the number of generations (as a proxy for age) instead of the number of years produces results more aligned with our hypothesis that dispersal modulates the effect of age on range size. We find that the effect of species age on range size decreases with an increase in body size and that species with aerial dispersal (bats), on average, have larger ranges than terrestrial species.

We included this in the main text in results Lines 252-266, Figure 6, in discussion in Lines 284-286 and 339-344, and methods lines 477-478, 524-531. Overall, our additional analyses suggest that using species age expressed in the number of years instead of the number of generations produces equivalent results, supporting our main conclusion that species age affects range size. However, using species age has the advantage of having more complete data across taxa, since generation time information is unavailable for most species in several taxonomic groups. Therefore, we still use the species age metric expressed in the number of years in the main analyses but now discuss the (lack of large) implications of using other metrics (Lines 284-286 and 339-344).

5. The absence of age-area relationships due to geographic constraints for islands is obvious and intuitive (which is related to the response variable, range size, and authors discuss other problems there), but the other interpretations of the variables and their relationship may be more challenging.

R5. From our understanding, the reviewer would like to see more in-depth interpretations on why insular species show less strong species age- range size relationships, besides due to geographic constraints. Note that we have modified this section because our results have changed (as explained in the first comment). We found actually a positive relationship between age and range size for all groups. We have included a new discussion section in lines 304-333.

6. Finally, although authors discuss the taxonomic issues, it would be nice to be more explicit about the latitudinal taxonomic gradients more recently discussed by Freeman and Pennell (2021, <https://doi.org/10.1016/j.tree.2021.05.003>).

R6. Thanks for the suggestion. We have explicitly included the latitudinal taxonomic bias and its implications in the discussion section lines 387-389: “Moreover, taxonomic classification biases are latitudinally dependent, where underestimating the number of species in the tropics might result in overestimating range and age (Freeman and Pennell, 2021).”

7. The above problems are quite resistant to solutions for broad-scale, interspecific studies as proposed in this paper. One of the assumptions is that all these confounding factors at the macroscale may

be random errors and increase the noise in data, but even so, the main trends would be potentially detectable. In this sense, it is quite common that these macroecological relationships are better viewed as constraint envelopes rather than overall functional (linear or exponential) relationships. Moreover, the above issues most likely produce bias whose directions are hardly predictable. Indeed, as can be seen in figures 1 and 3 of the manuscript, the effect size is TOO low.

R7. We appreciate the reviewer's comments and agree that confounding factors, some of which we have now evaluated in sensitivity analyses (see previous responses), increase noise in macroecological data. However, we also note that main trends can still be detected at this scale, even when effect sizes appear small. The reviewer questions this general trend, noting the low effect sizes in our age-range size relationship. To rigorously assess this, we built a Null Model in which species ages were randomized across taxa 1,000 times. Each randomization was analyzed through linear mixed-effects models to test the age-range size relationship (yielding 1,000 models). The histogram below shows the 1,000 model estimates for each taxonomic group, with the empirical data's estimated value denoted by a vertical red line. When the phylogenetic signal of species ages is broken, the species age effect on range size centers around zero (neutral effect), showing no significant deviation from the observed values for non-significant groups (e.g., marine mammals). However, the groups that empirically show a significant effect of age on range size (reef fishes, palms, reptiles, terrestrial mammals, amphibians, and birds) differ substantially from this random scenario. This sensitivity analysis, added to the main results in lines 125-128 and Figure 2, discussion in lines 282-284, and methods in lines 489-494, demonstrates that despite small effect sizes, they are much larger than those expected by chance.

Additionally, the reviewer raises points about inherent limitations in broad-scale, interspecific studies. While such studies have challenges, as we acknowledge below, they offer essential insights that are often unattainable in smaller-scale studies, which face even greater issues, including limited generalizability and statistical power (see, e.g., Nature, S41559-023-02144-3). Furthermore, the observed effect sizes in our study—around 0.15 for many species groups—are substantial compared to values typically reported in ecological research. For example, average effect sizes in ecological observational studies are often lower, with weighted means around 0.05, unweighted medians at 0.15, and weighted means of experimental studies at 0.19. Thus, our effect sizes are consistent with those commonly encountered in ecology, reinforcing the robustness of our findings.

Frequency distribution of effect sizes of age on range size from linear mixed-effect models based on 1,000 randomizations of age across the phylogeny for all taxa combined and each taxonomic group (indicated by blue bars). The red line denotes the observed effect size of age on range size for the empirical data.

8. Following the issues raised about the low effect size revealed in Figures 1 and 3, it is important that, although there may be statistical significance effects, there are potentially inflated Type I errors due to phylogenetic autocorrelation, and thus, I'm afraid that most of the coefficients would be statistically closer to zero than shown. It is unclear to me that phylogenetic structure was accounted for in the mixed model, and thus, what is the meaning of the P-values shown throughout the manuscript.

R8. We corrected for phylogenetic non-independence for all analyses (including Fig. 1 and 4) by including a phylogenetic taxonomic structure in the random effects (Family nested within Order nested within taxonomic group when models allowed it). Although a PGLS is ideal, running such a model including all +26,000 species with six independent phylogenetic trees is computationally too demanding. Nevertheless, as shown by Soul et al. 2015 (<https://doi.org/10.1093/sysbio/syv015>), accounting for taxonomic structure in mixed models (in the random effect) can be as good as explicitly accounting for phylogenetic dependence (e.g., PGLS). We have clarified this in methods lines 484-485, 504, 520-523.

9. I'm not saying that all the above issues would completely jeopardize the manuscript's conclusions, and I applaud the authors' efforts to attack this important problem. At the present time, I'm a bit skeptical about the result, but if other reviewers and authors are confident about their findings, I have no hard feelings about this.

R9. We hope our responses and the new sensitivity analyses have addressed the doubts. As the reviewer noted, we also recognize this is a complex question and study with many nuances. We have done our best to acknowledge these nuances while remaining within the current methodologies' limits.

Reviewer #2 (Remarks to the Author):

10. This manuscript demonstrates the positive relationship between the evolutionary age of species and their range size. The conclusions drawn from the results seem valid and I detect no serious methodological issues. The relationships are not strong, but this is discussed, and they are based on a lot of data, across many species. Overall, I think this manuscript is an interesting and useful contribution. I have a few suggestions for clarifications.

R10. Thanks for the positive feedback and the suggestions.

Suggested improvements and points to address:

11. The title is a bit complex.

R11. We have changed the title to “Evolutionary age influences range size across plants and animals.”

12. Lines 56-59: Please provide references that directly address these important arguments or convey their uncertainty. These arguments are repeated throughout the manuscript and therefore need a good foundation.

R12. We have included relevant references in lines 56-59: “Species with narrow geographical ranges face a higher risk of extinction compared to widespread species¹⁻⁴. Narrow-ranged species tend to have lower overall abundance and smaller local populations⁵⁻⁷, making them vulnerable to environmental perturbations that result in local extinction².”

13. Not all island species are restricted to one island, their range size can cover many islands. Additionally, some species occur both on islands and on mainland. Therefore island-dwelling is not necessarily equal to being restricted to a small range. Please discuss how this affects the assumptions and conclusions of this study.

R13. Thanks for pointing this out. We now clarify this in the method section lines 457-470 : “We collated data on insularity (land masses smaller than Australia) for 26,784 species. We classified species as “restricted to islands” if found exclusively on islands. Conversely, species “not restricted to islands” are those found on continents or on both continents and islands. Despite some island endemic species inhabiting multiple islands, their maximum potential range is still generally less than that of continental species or species that live both on continents and islands. We obtained data on island endemism from Phylacine 1.2.1^{75,76} for terrestrial mammals. We considered bird species as “restricted to islands” when they were reported to be 100% associated with islands in Sheard et al.⁸². For amphibians, we obtained information on insularity by overlaying species distribution maps with a shapefile of islands of the world, which is based on a digital elevation model at 30 m of pixel resolution. For reptiles, we obtained information on geographical context (“restricted to islands” vs “not restricted to islands”) from Meiri⁷⁹, and for reef fishes from Alzate and colleagues⁸⁰, Robertson & van Tassell⁶⁷ and Robertson & Allen⁶⁸. We classified palm species as restricted or not restricted to islands following Cassia-Silva et al.⁸³”

While it is true that some island species are not restricted to a single island and that some occur both on islands and the mainland, these distinctions do not alter this study's main assumptions or conclusions. Our classification focuses on distinguishing species based on their geographical exclusivity to islands versus a broader range that includes continents. By treating island endemics as having smaller potential ranges relative to non-restricted species, we aim to capture broad patterns of range size variation. The overall conclusions, particularly regarding differences in range size and potential ecological or evolutionary implications, remain robust to these definitions. However, we acknowledge that additional granularity, such as explicitly accounting for species inhabiting multiple islands or both island and mainland habitats, could

further refine future analyses. Unfortunately, these data are currently not available for all taxa included in our study.

14. Lines 91-101: Consider clarifying this long sentence by splitting it.

R14. We have split the sentence and restructured the paragraph in Lines 91-112: “Here, we determine the relationship between species age and range size for over 26,000 species from across the Tree of Life, including marine and terrestrial mammals, birds, amphibians, reptiles, reef fishes, and plants. We hypothesize that species age has a positive effect on range size. The strength of this effect may depend on taxonomic scale, geographical context and species' dispersal abilities. First, we expect the relationship between age and range size to be more pronounced at large taxonomic scales (e.g., orders) than at narrow taxonomic scales (e.g., families). Range size and/or age variability tend to be lower at smaller taxonomic scales consisting of closely related species that share similar ages, distributions, and dispersal abilities, leading to no or weak age-range relationships. Second, we hypothesize that the relationship between age and range size is less pronounced or absent on islands than the mainland, because the maximum range size that endemic species can attain on islands is geographically rather than age-constrained. Third, we expect that the relationship between age and range size is more pronounced for less dispersive than highly dispersive species, because dispersive species may attain maximum ranges faster than less dispersive species, reducing the effect of age on range size. Hence, dispersive species may have larger ranges, and less dispersive species smaller ranges, than expected based on age only. Our results show that age positively affects range sizes, but this relationship is influenced by taxonomic scale, geographical context, and the species' dispersal abilities. Our study reveals that species with small ranges, and thus increased extinction risk, tend to be restricted to islands, are poor dispersers, and/or are younger. Understanding these dynamics is crucial for predicting species' vulnerability to extinction, especially in the context of changing environmental conditions and the need for targeted conservation efforts.”

15. Lines 77-88 and 157-173: Using the broad description of “geographical constraints” seems inappropriate because not all types of geographical constraints (e.g. mountains) have been tested here and this study instead only refers to island vs. non-island species.

R15. We agree. Thanks for pointing this out. We emphasize more that we refer to geographical constraints imposed by islands in lines 99-102, 183-185, 204-206, 304-309, and 457-470.

16. A clear definition of what is meant by a dispersive species is lacking in the introduction but is needed here already to understand the arguments.

R16. We have modified the sentence to include a more explicit definition of dispersive species in lines 82-84: “Good dispersers (i.e., species that can move easily across barriers and/or over great distances) may attain large ranges faster than less dispersive species²². ”

17. Hypotheses 1 and 2 are very similar and are addressed together in the first results section. Other hypotheses have their own results sections. This can be confusing to the reader. Consider combining the first two hypotheses or splitting the first results section or another solution for clarification.

R17. Thanks for the suggestion. We have split the first section of the results into two sections: 1) Species age positively affects range size, and 2) The age-range size relationship varies across taxonomic scales.

18. Lines 113-114: “These results were robust to outliers”. It seems important to highlight this more (e.g. in figure caption) because the effect of outliers does come to mind when looking at the figures. The additional outlier test makes the results convincing.

R18. Thanks for the suggestion. We have expanded the information in the caption: “These results were robust to outliers and different approaches to calculating species age (Supplementary Fig. 1-2, Supplementary Table 3-4).”. We also expanded the results section in lines 121-128: “These results were robust to outliers (age values deviating more than three standard deviations from the mean, Supplementary Fig. 1, Supplementary Table 3) and different approaches of calculating species age²³ (e.g., when adjusting for extinction probability, Supplementary Table 4 and Supplementary Figure 2). Finally, null models (1,000 randomizations of age across taxa) that break the phylogenetic signal of age supported our findings by showing that the observed effects (i.e., standardized effect sizes) of age on range size strongly deviated from a null expectation of no effect of age on range size for all taxa, except for marine mammals (Fig. 2).”

19. Lines 164-169 and Figure 3: The use of dashed lines to indicate both significant and non-significant relationships is confusing.

R19. Thanks for pointing this out. As all relationships are significant, we have now made continuous lines.

20. Lines 216-320: There is no header for the discussion.

R20. Thanks, we have included the header.

21. Lines 267-273: More nuance is needed in the discussion about the choice of proxies for dispersal ability. Large body mass or large fruit size is not always an indicator of high dispersal ability, small animals (birds) and small fruits (eaten by large animals or by bats that fly far) can also disperse over long distances.

R21. Thanks for the suggestion. We have expanded the discussion by including these aspects in lines 368-373: “While traits that confer high dispersal abilities, such as large body size, may allow species to attain larger ranges, the specific dispersal trait relationship to range size may be more complex and clade-dependent. The selection of relevant and meaningful dispersal traits is crucial, as it can significantly influence the effect sizes observed in the pattern of interest. Sometimes, a combination of traits better reflects the overall dispersal capacity of an organism²².”

22. Lines 306-309: “species with smaller ranges are more often threatened with extinction”. This seems to be a new result presented in the discussion. I cannot find the description of this in the supplementary methods. It is therefore unclear to me what has been done here. Please be aware that many IUCN statuses have been assigned based directly on the range size (Gaston & Fuller 2009, *Journal of applied ecology*). Therefore, any tests of relationships between range size and IUCN status would not be useful.

R22. We rephrased this paragraph as this is not a direct result of our study in lines 407-429: “Our study determined the effect of species’ evolutionary age and its interaction with insularity and dispersal in shaping range size variation. We show that species with smaller ranges, which are often linked to heightened extinction risk, are disproportionately restricted to islands, tend to have lower dispersal capacities, and are frequently younger. This limited range size is a critical factor explaining why island endemic species are disproportionately endangered or have recently gone extinct³¹. The evolutionary loss of dispersal abilities,

a phenomenon commonly observed in insular species, further exacerbates their extinction risk by constraining their ability to expand their ranges and respond to environmental changes^{37,62}. Islands, while often biodiversity hotspots, can act as eco-evolutionary "traps," where the combination of restricted ranges, physical barriers to dispersal, evolutionary loss of dispersal traits, and local extinctions intensifies the risk of lineage extinction. While older species typically have larger ranges, many exceptions exist, such as "living fossils" that are species threatened by extinction because of being confined to small ranges despite their evolutionary longevity⁶³. By integrating geographical context, clade-specific traits, and phylogenetic history (species age), we provide a more nuanced framework for assessing extinction risk. There is a need for targeted conservation efforts, particularly for island ecosystems and range-restricted species disproportionately impacted by natural and human-driven threats. Ultimately, our study sheds light on how ecological, evolutionary, and geographical processes might simultaneously shape species' vulnerability to extinction. Understanding these dynamics is critical for predicting future biodiversity loss and informing conservation strategies that safeguard the persistence of Earth's species in an era of unprecedented environmental change."

23. Lines 347-356: Please clarify if this criterion is true for all groups: "We considered bird species as "restricted to islands" when they were reported to be 100% associated with islands". How were species categorized that occur both on mainland and islands?

R23. We have clarified this in lines 456-459: "We collated data on insularity (land masses smaller than Australia) for 26,784 species. We classified species as "restricted to islands" if found exclusively on islands. Conversely, species "not restricted to islands" are those found on continents or on both continents and islands."

24. Lines 357-359: Please indicate what this is used for. Is it to select species as described in lines 324-327?

R24. This has been removed.

25. Please provide more clarity earlier on in the manuscript about what the dispersal related traits are exactly.

R25. We included a definition in lines 219-222: "Dispersal-related traits are an organism's attributes associated with movement, endurance, or colonization that may enhance dispersal²⁵. As dispersal-related traits, we selected body size (for mammals, reef fish, amphibians and reptiles), egg type (for reef fishes), flight ability (for mammals), hand-wing index (for birds) and fruit size (for palms, see methods)."

26. A discussion is missing of the possible influence of a difference between present-day range size and range size before human disturbance. This seems crucial.

R26. Thanks for pointing this out. We now include information on the effect of human-driven contractions and extinction in lines 329-332: "Contrary to expectations, older palm species on islands do not exhibit larger ranges. This anomaly may be attributed to anthropogenic factors, such as habitat transformation, that have led to range size reductions of evolutionary old (and possibly phylogenetically unique - i.e., descending from long branches) palms.", lines 357-360: "Possibly, the Quaternary extinctions and ongoing declines of many large-bodied animals (megafauna) have led to dispersal limitation of palm species with

large fruits, leading to local extinctions and range reductions ^{45,46}, and the recovery of a positive age-range size relationship.” and lines 381-386: “Second, human-induced range contractions and extinction, such as the ones occurring during the Quaternary but also nowadays, have particularly affected ranges of large-bodied animals and indirectly range sizes of plants that depend on them for dispersal ⁵³⁻⁵⁶. The detection of recently diverged or near-extinct species and the accurate estimation of range size, is impacted by unnatural (e.g., human-driven) extinction and extirpation events, and geographic sampling biases ^{13,48,57}.”

Response to Reviewer's Comments

Reviewer #1: Remarks on Code Availability

Response: Thank you for your comment. We have now included the R code used for the analysis, which is indicated in the **Code availability** section.

Reviewer #2: General Remarks

Comment 1: After carefully reading through the new manuscript and the responses to the reviewer comments, I have no further questions. I think all points have been addressed well, the methods are clearer, and the discussion is now more comprehensive.

Response: We greatly appreciate your positive feedback and are pleased that the revisions have clarified the methods and improved the discussion.

Comment 2: Are the R code formulas used for the linear mixed-effects models written out somewhere (supplement)? It is a bit unclear otherwise how the nestedness levels were done technically.

Response: We have now included the R code used for the analysis, which is indicated in the **Code availability** section. An example of the formula for the linear mixed-effects models, including the nested random effects, is as follows:

```
BaseModel_rand <- lmer(Ranges ~ Ages + (1 | Groups / Order / Family), data = RangeData)
```